# From Hematopoietic Stem Cells to Platelets: Unifying Differentiation Pathways Identified by Lineage Tracing Mouse Models

**DOI:** 10.3390/cells13080704

**Published:** 2024-04-19

**Authors:** Bryce A. Manso, Alessandra Rodriguez y Baena, E. Camilla Forsberg

**Affiliations:** 1Institute for the Biology of Stem Cells, University of California-Santa Cruz, Santa Cruz, CA 95064, USA; 2Department of Biomolecular Engineering, University of California-Santa Cruz, Santa Cruz, CA 95064, USA; 3Program in Biomedical Sciences and Engineering, Department of Molecular, Cell, and Developmental Biology, University of California-Santa Cruz, Santa Cruz, CA 95064, USA

**Keywords:** hematopoietic stem cell (HSC), megakaryopoiesis, thrombopoiesis, megakaryocyte progenitor (MkP), platelet, lineage tracing, transplantation

## Abstract

Platelets are the terminal progeny of megakaryocytes, primarily produced in the bone marrow, and play critical roles in blood homeostasis, clotting, and wound healing. Traditionally, megakaryocytes and platelets are thought to arise from multipotent hematopoietic stem cells (HSCs) via multiple discrete progenitor populations with successive, lineage-restricting differentiation steps. However, this view has recently been challenged by studies suggesting that (1) some HSC clones are biased and/or restricted to the platelet lineage, (2) not all platelet generation follows the “canonical” megakaryocytic differentiation path of hematopoiesis, and (3) platelet output is the default program of steady-state hematopoiesis. Here, we specifically investigate the evidence that in vivo lineage tracing studies provide for the route(s) of platelet generation and investigate the involvement of various intermediate progenitor cell populations. We further identify the challenges that need to be overcome that are required to determine the presence, role, and kinetics of these possible alternate pathways.

## 1. Introduction

The product of the megakaryocytic lineage are platelets, a blood component absolutely required for life [1,2]. Platelets (thrombocytes) are short-lived, small, anucleate cell fragments that arise via shedding into the blood from proplatelet cytoskeleton extensions from their parent cell, the large polyploid megakaryocyte, and/or by megakaryocyte budding or fragmentation (thrombopoiesis). Primarily involved in hemostasis [1,3], platelets also exhibit other functions related to immunity and cell communication depending on their local microenvironment (reviewed here [3]). The clinical relevance of platelets cannot be understated [4,5,6] and, as such, understanding the developmental pathway(s) leading to their formation may reveal therapeutic targets to prevent and/or correct adverse thrombotic events, such as venous thrombosis, thrombocytopenia, thrombocytosis, and ischemic stroke (reviewed here [7,8,9,10,11]).

Adult mammalian bone marrow (BM) is host to the formation, maturation, and residence of megakaryocytes [2]. In classical models of hematopoiesis [12,13] (Figure 1), platelets have long been described as arising via the successively lineage-restricting differentiation of hematopoietic stem cells (HSCs), which reside at the apex of the hematopoietic hierarchy. HSCs differentiate into multipotent progenitor cells (MPPs), which no longer self-renew yet maintain multipotency [14,15]. Further developmental progression through the oligopotent common myeloid progenitor (CMP) [16] occurs prior to the transition to the bipotent megakaryocyte–erythroid progenitor (MEP) before commitment to a unilineage megakaryocyte progenitor (MkP) [17,18,19]. After this stage, several maturation steps occur where MkPs develop into megakaryocytes by undergoing molecular and cellular changes, including cytoplasmic remodeling and increasing in size and ploidy, before fragmenting and/or forming proplatelet extensions from which platelets shed into the blood circulation [1,3]. Platelet generation (collectively, megakaryopoiesis and thrombopoiesis) is thought to largely parallel erythropoiesis (red blood cell/erythrocyte formation) due to this well-accepted model that positions both lineages immediately downstream of the bipotent MEP, necessitating shared progenitor populations for much of their differentiation trajectory [17,20]. However, this traditionally accepted view of megakaryocytic specification is being contested [21], both at steady-state and under stress, highlighting the need to specifically and accurately trace the cellular origin(s) of platelets in situ, undisturbed, and at the single-cell level. 

One powerful approach for studying cell differentiation pathways, including platelets, is lineage tracing, which is used to recreate partial or complete cellular lineage trees. First pioneered by Charles O. Whitman in 1905 [22], the earliest instances of lineage tracing involved the microinjection of dyes into cells and tracking their progeny [23,24]. Over the past century, this technique has adopted newer and more powerful technologies driven by the development of chimeric mice [25], genetically driven fluorescent reporter-based systems [26,27,28], cellular/DNA barcoding [29,30,31], CRISPR/Cas9 scarring [32,33,34,35,36,37,38,39], the identification and tracking of naturally occurring somatic mutations [40,41,42,43,44,45,46,47,48,49], and complex combinatorial approaches integrating two or more of these systems [50,51,52,53,54]. Importantly, most modern approaches make use of a permanent mark in a parental cell (such as HSCs) that is inherited by all daughter cells and their progeny. Lineage tracing has been widely applied and incredibly impactful in understanding HSCs and hematopoiesis [55], including seminal studies [56,57,58,59,60,61,62,63,64] that revealed the functional and differentiation cornerstones of HSCs that informed early iterations of the classical hematopoietic hierarchy [16,65]. Although outside the scope of this review, we acknowledge that recent lineage tracing data have refined how the field views HSC differentiation and hematopoiesis across multiple lineages. However, even though lineage tracing has markedly increased in resolution, sensitivity, and applicability, significant challenges remain when interrogating specific cell types, including those of the megakaryocytic lineage. 

Many of the classical hematopoietic differentiation steps in platelet generation have been inferred via transplantation, a highly useful technique yet one that is likely to reflect stress and/or high-demand physiological states due to the need to precondition the host (such as with irradiation or other treatment). However, given the clinical and life-saving significance of transplantation, understanding platelet development beyond steady-state could offer important insights for clinical applications. Thus, even though transplantation studies have elucidated much of our current knowledge, the most direct evidence for the developmental path of steady-state platelet formation comes from undisturbed in situ lineage tracing.

There have been numerous studies of lineage tracing from HSCs, but few have assessed the megakaryocytic/platelet lineage largely due to platelets being devoid of a nucleus, thus lacking the active expression of reporter genes and making them unsuitable for lineage tracing via genetic barcoding or scarring, although new approaches may overcome this challenge [66]. Furthermore, in the context of transplantation studies, platelets lack expression of the pan-hematopoietic marker CD45, further limiting the tools available to track differentiation dynamics. Additionally, HSCs and MkPs exhibit striking phenotypic and molecular similarities (reviewed here [67]). These limitations substantially reduce the number of tools available to lineage trace platelets. Additionally, among the few lineage tracing studies that have directly assessed platelets, only a small number have investigated the intermediate progenitors between HSCs and platelets in the BM. Here, we specifically review lineage tracing data that seek to understand the cellular origins and progenitors of platelets in adult murine BM at steady-state, including recent data suggesting alternative routes of generation, and highlight the successes and challenges inherent to these models. We conclude by discussing the current feasibility of megakaryocytic-specific lineage tracing.

## 2. HSCs Produce Platelets, but Do Some HSCs Do It Better Than Others?

In the adult mouse, transplantation studies of labeled HSCs, including quantitative analysis by our group [20], revealed that platelets (and all other blood lineages) are effectively reconstituted following HSC engraftment, indicating their hematopoietic (and HSC) origin [68,69,70,71]. Additionally, recent evidence suggests that a subset of HSCs exist along a continuum of platelet bias, lineage priming, and/or restriction [68,70,72], similar to data suggesting heterogeneous HSC clonality and unilineage restriction of other cell lineages [15,73,74,75,76,77,78,79]. In this section, we explore key lineage-tracing studies that allow for direct observation of the platelet lineage.

Using a von Willebrand Factor (*vWF*)–GFP reporter mouse, Sanjuan-Pla et al. found that a subset of HSCs was labeled with GFP in situ [70], which was a surprising finding as vWF was previously reported to only be expressed in mature megakaryocytes, platelets, and endothelial cells. Functional analysis by transplantation of vWF+ (GFP+) and vWF− (GFP−) HSCs revealed that the vWF+ compartment produced more platelets and fewer lymphocytes than the vWF- fraction. Importantly, even though a bias was observed, both HSC subsets maintained multipotency. Additionally, vWF+ HSCs could give rise to both vWF+ and vWF− HSCs following transplantation, but vWF− HSCs never gave rise to their vWF+ counterparts, potentially indicating that vWF+ HSCs represent a cell state slightly further up the hierarchy (within the phenotypic HSC pool). Similarly, a follow-up study driven by Carrelha et al. refined this notion using a dual-color *Gata-1*-GFP and *vWF*-Tomato (Tom) lineage tracing model that allowed the specific assessment of platelet (and other blood cell) reconstitution [68]. Single vWF+ HSCs were transplanted into more than 1000 recipient mice, and the resulting lineage contribution was measured. Building on the original results obtained by Sanjuan-Pla, this study found that ~11–12% of vWF+ HSCs exclusively reconstituted platelets and no other lineage. Additionally, the data demonstrated a non-random lineage hierarchy of HSC clonal reconstitution capacity where transplanted vWF+ HSCs preferentially established platelet, then platelet-erythroid, platelet–erythroid–myeloid, and finally, platelet–erythroid–myeloid–lymphoid lineages. The finding that platelet generation from HSCs had the fastest kinetics (detectable donor-derived chimerism) was also observed by others [69,71] and may indicate a physiological mechanism for continual replenishment of this short-lived, critical blood component. Importantly, the secondary transplantation of platelet-restricted HSCs by Carrelha et al. revealed preservation of the platelet bias by some HSC clones, whereas others underwent multilineage reconstitution, as previously observed by others (and different models) [15,70]. These studies identified two HSC subsets newly defined by vWF expression, indicating that platelet generation is a shared feature of all reconstituted mice, and argued for the existence of megakaryocytic lineage bias in the phenotypic vWF+ HSC pool. The authors also argue against the notion of a unilineage megakaryocyte progenitor contaminating the vWF+ HSC pool as, upon isolation from recipient mice, vWF+ Lin−Sca1+cKit+ (LSK) cells showed robust multilineage capacity in vitro, indicating that, when combined with the secondary transplants, vWF+ HSCs are bona fide HSCs. However, one caveat to these (and all other related) studies is the use of transplantation to test the functional output of the labeled HSCs; it is possible that the observed results (summarized in Figure 2A) are more a measure of functional capacity upon stress rather than in situ, steady-state output. This acknowledgment is not meant to diminish these findings, as they are highly informative and clinically relevant, but rather to ask if complementary in situ approaches reveal key physiological differences.

To that point, a recent in situ lineage tracing study by Rodriguez-Fraticelli et al. utilized the doxycycline-inducible Sleeping Beauty transposon system and concluded that, at steady-state, over 30% of megakaryocytes can be directly derived from long-term (LT)-HSCs without contributions to other hematopoietic lineages [80], potentially indicating HSC lineage bias or restriction. However, this study did not extensively evaluate many other intermediate progenitor cell types, obfuscating the differentiation path each HSC clone utilized. Yet, the finding that some HSC clone barcodes were only shared between MPPs and MkPs, but not other myeloid progenitors, is intriguing. Furthermore, upon the transplantation of labeled LT-HSCs, multilineage reconstitution was observed, highlighting that most platelet-biased LT-HSC clones retain full multilineage capacity and that transplantation can result in discordant results compared to in situ analysis [80]. Such differences within a study or between studies and the conclusions drawn may also be attributed to the type of transplant (single cell vs. bulk) and the type of label utilized (individual clones labeled differently vs. a subset of HSCs containing the same label) (Figure 3). Additionally, confirming previous reports [15,70,81,82], Rodriguez-Fraticelli’s study demonstrates that a subset of the total LT-HSC pool exhibited a shift toward a megakaryocytic transcriptomic profile yet maintained multilineage reconstitution upon transplantation, further reinforcing the hypothesis of lineage bias over restriction.

Given the relatively short-term (up to 8 weeks post-label induction) tracing of the above studies, in situ evaluation over longer periods of time may refine the observation of potential platelet bias and the kinetics of lineage output by measuring recovered (near steady-state) blood cell reconstitution. One such study used two independent HSC inducible lineage tracing models, *Krt18*-CreER/YFP and *Fgd5*-CreER/Tom, and chased mice for one year post-label induction by tamoxifen administration [83]. Similar to the previous studies [68,69,70,71], they found that platelets showed the highest labeling efficiency (other than HSCs) early on, indicating a potential preference for and rapid kinetics of this lineage. Over time, all other lineages exhibited increasing, yet varying, levels of labeling. These findings were largely recapitulated by Morcos et al., who utilized a related *Fgd5*/zsGreen-CreERT2/RFP model with up to 92 weeks of chase post-tamoxifen label induction [69]. Another study, also taking advantage of the *Fgd5*-CreER/Tom model, induced labeling and was chased for up to 83 weeks [71]. In contrast to others, they observed initial labeling among HSCs and early progenitor cells. However, by 4 weeks post-label induction, platelets were the only mature cell type to express the induced label. Thus, it appears that HSCs, or at least a subset of HSCs sensitive to label induction due to Fgd5 expression, exhibit faster reconstitution of the platelet/myeloid lineage rather than lymphoid and that platelets are robustly and continuously replenished by HSCs (summarized in Figure 2B).

Similar arguments for an HSC continuum of platelet bias, lineage priming, and/or restriction in humans have been made, yet lack as much direct lineage tracing evidence as demonstrated in mice. Taking advantage of whole genome sequencing and clonal mutation analysis in humans as a method of retrospective lineage tracing, Osorio et al. concluded that there is platelet lineage bias in humans [84]. In this system, clonal somatic mutations were used to reconstruct lineage relationships, with lineages sharing similar mutational patterns assumed to be more related than lineages with a low level of mutational overlap. They reasoned that the unique mutational identity of megakaryocytes (which was different from all other blood cell lineages) indicated an earlier divergence of the megakaryocyte lineage compared to all other myeloid/erythroid and lymphoid lineages, potentially indicating that a subset of HSCs primarily contributes to megakaryopoiesis in humans at steady-state. 

The above findings, combined with the phenotypic and molecular similarities between HSCs and MkPs [67], further reinforce the proposed paradigm shift in understanding megakaryopoiesis (Figure 2). The observed priming of HSCs may also contribute during times of perturbed hematopoiesis, such as inflammatory stress. Indeed, a fraction of phenotypic LT-HSCs, termed “stem-like MkPs”, express the classical megakaryocyte lineage marker CD41 and megakaryocyte-lineage mRNA transcripts, including *Cd42b* and *vWF* [81]. However, the translation of these transcripts is suppressed until activation, commonly via inflammation. Transplantation of LT-HSCs fractionated by CD41 expression confirmed multilineage output, except for those with the highest levels of CD41, which were only obtainable post-inflammatory insults [81]. This inflammation-induced CD41^hi^ phenotypic LT-HSC subpopulation exclusively, but transiently, produced platelets upon transplantation. The finding that a subpopulation of vWF+ LT-HSCs is biased to the platelet lineage was supported by in situ lineage tracing and suggests that steady-state platelet production is not exclusive to the CD41+ LT-HSC fraction, as shown by Rodriguez-Fraticelli et al. [80]. Thus, these stem-like MkPs could represent steady-state platelet-biased HSCs that rapidly lose HSC function and gain platelet restriction upon exposure to inflammation or an independent progenitor population found within the phenotypic HSC pool that is held in reserve and primed for emergency use for specific inflammatory states [81,85]. Collectively, HSCs contribute to megakaryopoiesis and possess physiological and stress-induced bias, which is likely to preserve blood levels of the critical-for-life platelet, yet questions remain regarding their exact dynamics. 

## 3. Does Megakaryopoiesis Transition through MPPs?

Using dual-color *Flk2*(*Flt3*)-Cre mT/mG (termed FlkSwitch) lineage-tracing mice [86,87,88,89,90,91,92,93], our group was able to ascertain if hematopoietic lineages progress through non-self-renewing, multipotent MPPs (defined by Flk2 expression) [86,91]. As *Flk2* is expressed during the transition from HSC to MPP and detected as early as the short term (ST)-HSC state, Cre recombinase leads to a permanent “switch” from Tom to GFP expression and all progenies of MPPs must, therefore, also express GFP. Thus, any progenitor cell, intermediate transitory cell state, or terminally differentiated cell that transitions through a *Flk2*+ stage at any point during its differentiation path irreversibly expresses GFP. Using this model, we found a high proportion of cells expressing GFP among MPPs (all subsets) and equivalent proportions of GFP-expressing cells among downstream lymphoid and myeloid (CMPs, MEPs, MkPs) progenitors. Importantly, all terminal blood cell progenies (lymphocytes, myeloid cells, erythrocytes, and platelets) also expressed GFP at the same high level as MPPs. This in situ model demonstrates that, regardless of any HSC bias/restriction to the platelet lineage, adult steady-state megakaryopoiesis transitions through a *Flk2*+ stage during its developmental trajectory. Importantly, when MPPs or other progenitor populations (including CMPs, granulocyte–monocyte progenitors [GMPs], and MEPs) were isolated and transplanted, the cellular output of each compartment was consistent with our lineage tracing data; MPPs maintained the ability to produce all mature cell types, including platelets, while losing self-renewal, and more committed progenitors transiently produced their classically expected terminal progeny [20]. The transplantation of single or limiting numbers of MPPs (and HSCs) also demonstrated multilineage capacity and the expected cellular intermediates for each major blood lineage.

Indeed, other lineage tracing studies also support megakaryopoiesis transitioning through an MPP stage [94]. Transplantation of vWF+ HSCs, even those that are platelet-restricted, results in robust label detection among MPP2 (LSK Flk2−CD150+CD48+) cells, which is a subpopulation contained within the MPP pool proposed to enrich for platelet production [68]. Similarly, other HSC-labeling lineage tracing models enrich for subsequent labeling of MPP2 cells [71,83,95] and, compared to other MPP subpopulations, the MPP2 population was found to contain megakaryocytic-specific clones [80] and to cluster transcriptionally with more lineage-committed megakaryocyte/erythroid progenitors [71]. These findings are supported by another study that argues for functional megakaryocyte lineage bias in the MPP2 population [96]. Thus, although not always assessed, platelet generation appears to transition through an MPP cell state.

## 4. Are CMPs and/or MEPs Required Intermediates in Platelet Generation?

Classically, the oligopotent, heterogenic progenitor pool of the myeloid lineage [97], CMPs, are downstream of MPPs (Figure 1). Further bifurcation via the differentiation of this progenitor pool gives rise to the GMP and MEP progenitor populations, the latter of which generates MkPs and erythroid progenitors, which are the proposed unipotent precursors of platelets and erythrocytes, respectively. However, given the evidence discussed above, is there a role for CMPs and/or MEPs in platelet generation?

The FlkSwitch lineage tracing mouse model our group utilizes [86,87,88,89,90,91,92,93] uniformly labels Flk2− CMPs and MEPs with GFP, indicating prior transition through a *Flk2*+ stage, which is consistent with them serving as developmental intermediates of platelets. Additionally, transplanted CMPs (and MEPs) transiently produce platelets [20]. Using the inducible HSC-selective *Pdzklip1*-CreER/Tom lineage tracing mouse model, Upadhaya et al. found that, one-week post-label induction, Lin−cKit+Sca1−CD150−CD41− myeloid progenitors (containing phenotypic CMPs) were label-negative, whereas MPP2 and MkP populations contained label-positive cells [95]. However, when the inducible HSC labeling *Krt18*-CreER/YFP or *Fgd5*-CreER/Tom models were used, different results were observed [83]. In these models, one-week post-induction, HSCs and hematopoietic progenitor cell 2 (HPC2s; similar to MPP2) harbored label-containing cells, whereas there was nearly no label found in downstream progenitors or platelets. By four weeks post-label induction, platelets, and most other progenitors showed varying frequencies of label-positive cells, obfuscating the route(s) of platelet generation at this time point. Säwen et al., also using the *Fgd5*-CreER/Tom model, obtained similar results in the first four weeks post-labeling [71]. However, they concluded that at earlier time points, MkPs acquired labeling with faster kinetics than any other CMP/MEP progenitor state. Together, these studies may indicate that at least the partial replenishment of platelets could bypass CMPs (and MEPs) or indicate fast differentiation kinetics not suitable for labeling by these models (Figure 2B). However, these data do not discount platelets arising via a megakaryopoiesis pathway consisting of multiple intermediate progenitors.

A bipotent MEP has been described in the mouse [16,98], but does lineage tracing implicate its involvement in megakaryopoiesis? When our group transplanted MEPs, platelet and erythrocyte production was detected in recipient mice [20]. Carrelha et al., using their dual-color *Gata-1*-GFP/*vWF*-Tom lineage tracing post-transplant of vWF+ HSCs model, identified that some, but not all, mice that exhibited platelet-restricted output also contained labeled MEPs [68]. However, the frequency of label-positive MEPs was low, so it is possible that the contribution by MEPs was underestimated in this system. Additionally, inducible HSC lineage tracing conducted by Upadhaya et al. showed that, early on (when HSC, MPP2, and MkP exhibited labeling but CMPs did not), there was no or minimal labeling among MEPs [95]. Similarly, the *Krt18*-CreER/YFP and *Fgd5*-CreER/Tom HSC lineage tracing models employed by Chapple et al. also found enriched early labeling in HSCs, HPC2s, and platelets with minimal labeling among MEPs (and CMPs) [83]. The in situ Sleeping Beauty lineage tracing model Rodriguez-Fraticelli et al. employed also did not label any MEPs and attributed this to the possibility that the MEP stage is too transient to be detected [80]. If so, the labeling of MEPs may be less robust than other progenitor cell states and reflect an important caveat of these lineage tracing studies—progenitor cell states must be utilized long enough to allow for detectable label expression. It is worth noting that significant heterogeneity among phenotypic human MEPs has been identified [99], and there is likely similar heterogeneity in mice. Thus, it is possible that phenotypic analyses routinely performed do not accurately capture and/or delineate between myeloerythroid progenitors in murine BM. Collectively, this set of data suggests that at least a fraction of platelet generation may bypass CMP/MEP intermediates but does not rule out the possibility of multiple routes of production with different output kinetics.

## 5. Can Platelets Arise Directly from HSCs by “Skipping” Intermediate Cell States?

One hypothesis to arise out of some of the models generated from the studies discussed above [68,69,70,71,80,81,83,95] is the possibility of a shortcut/bypass mechanism whereby megakaryocytic-restricted HSCs directly give rise to MkPs/platelets without progressing through MPPs or other classical intermediate cell states. The evidence for this is largely the result of the rate at which labeled cells accumulate in any given cellular compartment post-HSC labeling. Many of the studies highlighted above specifically indicate that platelets accumulate a higher proportion of labeled cells far faster than many other cell types, including their own “canonical” CMP and MEP progenitors, seemingly indicating a direct LT-HSC > MkP/platelet pathway. Morcos et al. even suggested that approximately 50% of steady-state platelet generation is derived via such a pathway, whereas the remaining proportion is derived via the classically viewed hematopoietic hierarchy [69]. However, and of fundamental importance, our FlkSwitch lineage tracing model [86,87,88,89,90,91,92,93] directly contradicts a direct LT-HSC > MkP/platelet (i.e., MPP “bypass”) pathway under true steady-state in young adult mice as (1) all platelets have excised the Tomato reporter (indicating transition through a Flk2+/MPP stage downstream of the LT-HSC) and (2) label switching is abundantly detected as early as the ST-HSC stage. Thus, if LT-HSCs bypass all other progenitors for direct platelet generation, retention of the Tomato label in platelets would be present at a greater frequency than what is measured among MPPs, which we did not observe.

One possible way to reconcile these data is a strong effect of HSC clonal restriction paired with accelerated kinetics, where certain HSC clones preferentially only give rise to platelets with such accelerated kinetics that label retention among intermediate progenitors is too transient to measure in inducible lineage tracing models. Other considerations are the time it takes to transition between cell states if clonal expansion occurs and if labeled cells are exhausted from a cellular pool upon differentiation. That is, if labeled progenitors differentiate without clonally expanding (or if all clonally expanded, labeled cells differentiate), then low-level labeling within a compartment may be misinterpreted. A comprehensive assessment of every hematopoietic stem and progenitor cell at multiple successive time points would need to be conducted to measure the kinetics of labeled progenitors. Furthermore, given that one megakaryocyte is predicted to give rise to up to thousands of platelets [100], the low-level labeling of upstream progenitors does not necessarily preclude a high proportion of labeling among terminally differentiated platelets, especially given that platelets generated prior to label induction turn over quickly. It is tempting to hypothesize that certain HSC clones transiently activate *Flk2* gene expression so quickly that cell surface protein is undetected (i.e., maintaining a *Flk2*− LT-HSC phenotype), allowing excision of the Tomato gene in our FlkSwitch mice. The resulting progeny would be GFP+, preventing direct determination of the existence of an LT-HSC > MkP/platelet pathway. However, this scenario is irreconcilable with our data, demonstrating that platelets, erythrocytes, and granulocyte/macrophages always show similar proportions of labeling in our model, thus suggesting that they all share the same progenitor cell state. Clearly, additional work is required to interpret these results and to gain a better understanding of all platelet production paths at steady-state and upon increased platelet demand, such as that induced in classical transplantation assays.

## 6. Are MkPs Unilineage Platelet Progenitors?

Since their initial characterization in 2003 [17] and refinement in 2007 [19], MkPs have largely been thought to be the unilineage progenitor immediately preceding megakaryocyte maturation, as in vitro experiments in those studies have concluded. Additionally, to our knowledge, no in situ lineage tracing directly targeting the MkP cell state has been performed (see the following section). However, we recently conducted transplantations of MkPs and tracked blood cell output in recipient mice [18]. Interestingly, in addition to transient platelet generation, a small burst of erythroid (and nominal GM) cell reconstitution was observed. Even in the context of transplantation, these new data, in combination with the studies discussed throughout, highlight the need for additional in situ analysis of all stages of megakaryopoiesis and continued refinement of progenitor cells. To that point, Säwen et al. compared the equilibrium ratios of MkP labeling to that of platelets and found that, in the *Fgd5*-CreER/Tom model, MkPs appear to be an obligatory step preceding platelet generation [71]. However, the phenotypic and functional heterogeneity of the MkP compartment has yet to be fully elucidated and might reveal additional plasticity given our MkP transplant data [18].

## 7. Is Megakaryocyte-Specific Lineage Tracing Possible?

We recognize that megakaryocytic lineage tracing using a lineage-specific gene and a Cre-based system can be challenging due to the established phenotypic and molecular similarities between HSCs and MkPs [67]. However, there are a few tools that have demonstrated potential success. The most commonly employed megakaryocyte lineage system is *Pf4*(*CXCL4*)-Cre, initially described by Tiedt et al. [101] and most commonly used for megakaryopoiesis-selective genetic deletion. This model was quickly adapted to fluorescent reporter lineage tracing studies [82]. However, the first example of this found that Pf4-Cre expression was not restricted to the megakaryocyte lineage but also labeled BM HSCs and their progeny between ~40 and 60%. This “non-specificity” could potentially strengthen the evidence for megakaryocyte lineage priming in HSCs. In this study, Pf4-Cre activity was found in ~41% of phenotypic LT-HSCs, with varying levels in the Lin−cKit+Sca1+ (~55%) and Lin−cKit+Sca1− (~53%) compartments, broadly comprising HSCs/MPPs and CMPs/GMPs/MEPs/MkPs, respectively. Label was also detected in a large frequency in lymphoid, granulocyte, erythroid, monocyte, and osteoclast cell types. This could be indicative of two potential conclusions: (1) some LT-HSCs express megakaryocytic lineage genes, potentially due to lineage priming or as a megakaryocyte-biased subpopulation, but clearly still maintain multilineage differentiation potential or (2) the *Pf4*-Cre system is not megakaryocyte-lineage-specific, and results generated using that model should be carefully considered for off-target labeling. Importantly, these two conclusions are not mutually exclusive. In contrast, others have found that the *Pf4*-Cre system is largely selective for the megakaryocyte lineage with little expression in other lineages, although non-megakaryocyte lineage labeling increased slightly during inflammation [102]. Together, these contradicting data indicate that the *Pf4*-Cre model may be useful for megakaryocyte lineage tracing but could potentially be hindered by suboptimal specificity.

Recently, an alternative system to Pf4-Cre was developed: the *Gp1ba* (*Cd42b*)-Cre mouse line [103]. As summarized in a recent review [104], the *Gp1ba*-Cre model has been characterized as acting later during megakaryopoiesis (primarily at the megakaryocyte) than *Pf4*-Cre but may be more selective for the megakaryocytic lineage. However, our recent bulk RNA-seq analysis indicates *Gp1ba* expression among MkPs [18]. It is, therefore, conceivable that a potential combination (i.e., dual reporter) of *Gp1ba* and a more upstream gene (such as *Pf4*) could be used for lineage tracing of late megakaryocyte development, maturation, and platelet generation. And, if *Pf4* is found to truly be expressed in HSCs, its combination may provide a unique system in which specific megakaryopoiesis cellular intermediates and events can be interrogated.

## 8. Can Discordant Lineage Tracing Outcomes Be Unified?

In this review, we have highlighted the major lineage tracing findings for platelet development. Further, we underscored the conclusions drawn based on the employment of either transplantation-based or in situ-based lineage tracing experimental design and explored the discordant results. However, is it possible to unify observations from both methods into an updated model of steady-state platelet generation? Given the critical need for platelets to sustain the life of an organism, it is not surprising that multiple redundancies and/or competencies may exist to ensure constant platelet production. This may help explain why some classically defined progenitor populations may be dispensable under some conditions and why at least a few HSC clones are primed for the megakaryocytic lineage. Further, a recent study sought to re-evaluate some of the lineage tracing models reviewed here to determine if diverse in situ HSC labeling strategies could be unified [105]. It was suggested that the seemingly discordant results obtained from different models (specifically *Fgd5*-CreER [71,83] and *Krt18*-CreER [83]) can actually be integrated mathematically, at least with respect to the labeling kinetics of LT-HSCs, ST-HSCs, and MPPs, as each strategy marks HSCs with slightly different properties. They further conclude that this reconciliation highlights the heterogeneity of HSCs and reinforces the importance of diverse experimental approaches and careful interpretation. It should, therefore, be possible to extrapolate this mathematical reinterpretation to evaluate the platelet lineage across multiple studies to potentially, if only predictively, unify the diverse findings of in situ lineage tracing. It is worth noting that two independent studies that used the same *Fgd5*-CreER/Tom model arrived at discordant conclusions [71,83], possibly indicating that different experimental approaches (i.e., tamoxifen administration strategies) should also be accounted for.

Given the interpretations of the data reviewed herein, we propose an updated, unified model of megakaryopoiesis (Figure 4). The heterogeneous HSC comportment comprises vWF^hi^ (and/or CD41^hi^) [68,70,80,81] LT-HSCs residing at the apex of our proposed hierarchy. Immediately downstream are vWF− LT-HSCs that give rise to a heterogeneous MPP pool. The majority of lineage tracing studies identified the MPP2 (HPC2) subset as the likely primary MPP intermediate involved in platelet generation [68,71,83,95], yet others may contribute as well. MPPs may then transition through the classical hierarchy of CMP > MEP > MkP or bypass CMPs and/or MEPs altogether. Specifically, MPP2 cells may comprise the subset of MPPs that directly or preferentially differentiate into MkPs, with specific HSC subsets (i.e., vWF+) feeding the MPP2 compartment. Convergence upon an MkP does appear to be an obligatory step in platelet generation. MkP specification and platelet differentiation may occur in a biased manner or operate in conjunction with the classical view of hematopoiesis. We propose that platelet output from HSCs is a “default” pathway of hematopoiesis [20] as they are absolutely critical for life, and thus multiple, non-exclusive routes can coexist. However, given the data reviewed here, we ascribe to a model whereby HSCs, at steady-state, give rise to platelets via the shortest possible path that still allows for the acquisition of alternative lineage output as demand requires (Figure 5).

## 9. Conclusions, Outlook, and Open Questions

Clearly, recent advances in lineage tracing have expanded the traditional view of megakaryopoiesis. However, even though multiple parallel routes of platelet generation are not necessarily mutually exclusive, future endeavors are required to confirm pathway(s), their importance and prevalence, and if they are altered or induced outside of steady-state megakaryopoiesis. This includes obtaining a better understanding of platelet generation path(s) during ontogeny and aging, following bleeding or hemorrhaging, in response to injury and infection, and in disorders such as autoimmunity, immune thrombocytopenia, and cancer.

Beyond their classically defined role in platelet production, megakaryocytes from both mice and humans have been further classified based on their transcriptome, cell surface phenotype, and function [106,107,108,109]. Three primary megakaryocyte subtypes have been identified: platelet-producing, BM niche-supporting, and those involved in immune and inflammatory responses. While we do not yet understand the plasticity between these megakaryocytic cell states, nor if aging or stress alters their specification or causes shifting between cell states, it will be interesting to investigate if specific route(s) of megakaryocyte specification (Figure 4) preferentially generate one or more subtypes. As an example, recent data from our group identified a parallel megakaryocytic pathway in aging FlkSwitch mice that arose directly from HSCs and generated MkPs with enhanced function compared to their classically derived or young counterparts (Poscablo et al. in press [110]). In this model, this age-unique pathway produced a subset of platelets that were hyperreactive, a state likely derived from their parent megakaryocyte. Thus, continued development of lineage tracing strategies is critical to address questions in route selection, megakaryocyte subtype specification, and resulting megakaryocyte and platelet function. 

Multiple challenges remain for the elucidation of the megakaryocytic lineage, including ensuring that labeling is robust enough to be detected in both progenitors and platelets, inducing highly selective megakaryocytic labeling, and addressing the clonality of HSCs, MkPs, and any intermediate progenitors. Additionally, inducing label expression experimentally (i.e., via tamoxifen administration such as commonly used in the studies reviewed here) may reflect stress and/or alter hematopoietic output that no longer fully recapitulates steady-state. These experimental challenges extend beyond model organisms, as the inability to retrospectively lineage trace human platelets via somatic mutations limits the opportunity to investigate platelet generation pathway(s) in humans. 

To that point, it is worth asking if studies in mice can directly inform human biology. Although worthy of its own dedicated and in-depth review, we offer the perspective that even if likely differences between species exist, the outcome is likely the same: platelets are required for life, and thus, multiple/redundant paths may be in place to ensure their continuous production. A relatively linear hematopoietic hierarchy has been the primary model for the human system (as in Figure 1), yet recent studies have challenged this view, proposing a more amorphous and plastic perspective of human HSCs and MPPs [111]. Furthermore, it has been suggested that the human megakaryocyte lineage diverges from HSCs earlier than other lineages, potentially similar to mice [112,113]. Comparative transcriptomics between human and mouse hematopoietic stem and progenitor cells reveals high levels of similarity with concordance between cell state hierarchies [114]. Some human HSCs/MPPs also demonstrate megakaryocytic lineage priming, the signatures of which are consistent with their murine counterparts [111,115]. Furthermore, upon the clinical transplantation of HSCs, platelets are the fastest recovering blood cell lineage [116], as highlighted within for mice [69,71]. Collectively, studies conducted in mice may translate well to human megakaryopoiesis, yet direct evidence is required to confirm any such conclusions.

An improved understanding of megakaryopoiesis has tremendous therapeutic potential by offering significant, clinically relevant cellular targets to modulate platelet generation. Furthermore, lineage tracing in any hematopoietic compartment, including megakaryocytes, could benefit from a host of improvements that include (1) refining the markers and strategies employed for HSC and progenitor cell identification, (2) discriminating between a lineage-committed cell, one that is lineage primed, and/or between multipotent cells located in an environment that is permissive to a single lineage [117], (3) the improvement and development of both in situ models and lineage tracing technologies [118], and (4) enhanced graphical representations [119].

We have previously proposed that erythrocyte and/or platelet generation may be the default fate of HSCs and hematopoiesis [20] (Figure 5). Indeed, the lineage tracing studies reviewed herein support the hypothesis of alternative, co-existing, continuous pathways supporting the critical-for-life generation of megakaryocytes and platelets. Thus, adult HSCs may primarily function to support blood integrity and oxygen transport via platelet and erythrocyte generation at the expense of immune cell replenishment [20,120].

## Figures and Tables

**Figure 1 cells-13-00704-f001:**
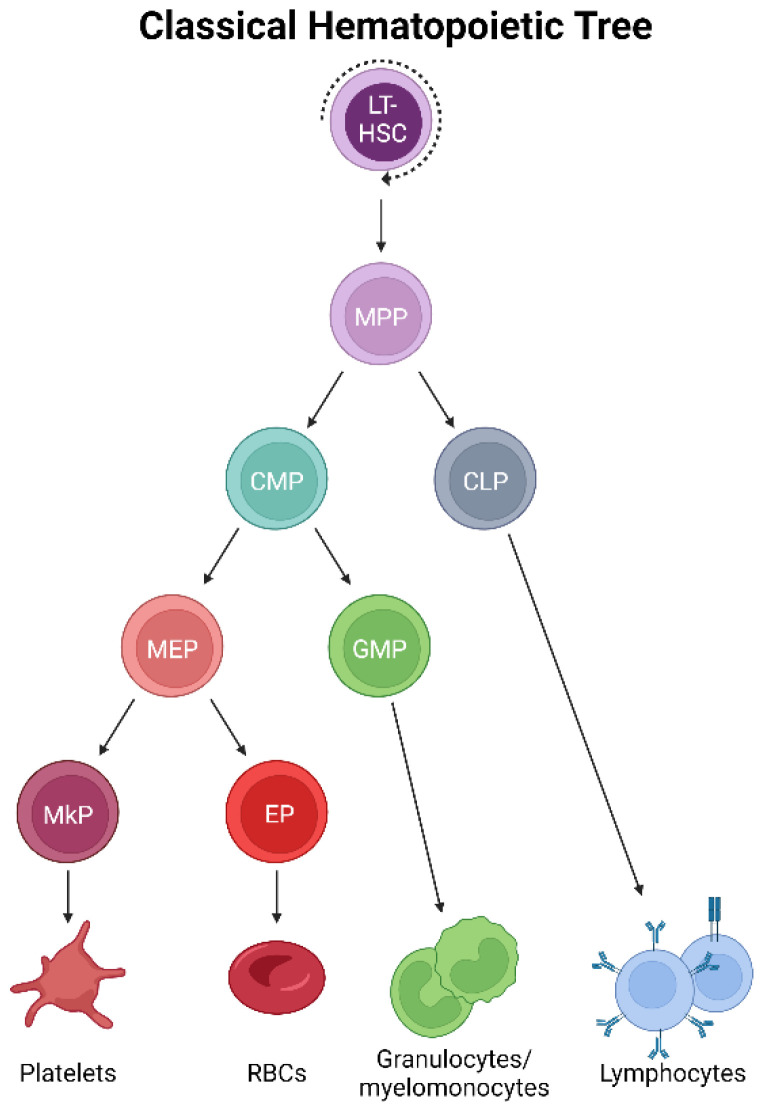
Classical hematopoietic tree. Self-renewing, multipotent HSCs reside at the apex of the hematopoietic hierarchy. The differentiation to MPPs results in the loss of self-renewal yet maintains multipotency. Successive differentiation then occurs, with downstream progenitor pools becoming progressively more lineage-restricted. Classically, platelets arise by the differentiation of MPPs into CMPs, MEPs, and MkPs, which mature into megakaryocytes that ultimately generate platelets.

**Figure 2 cells-13-00704-f002:**
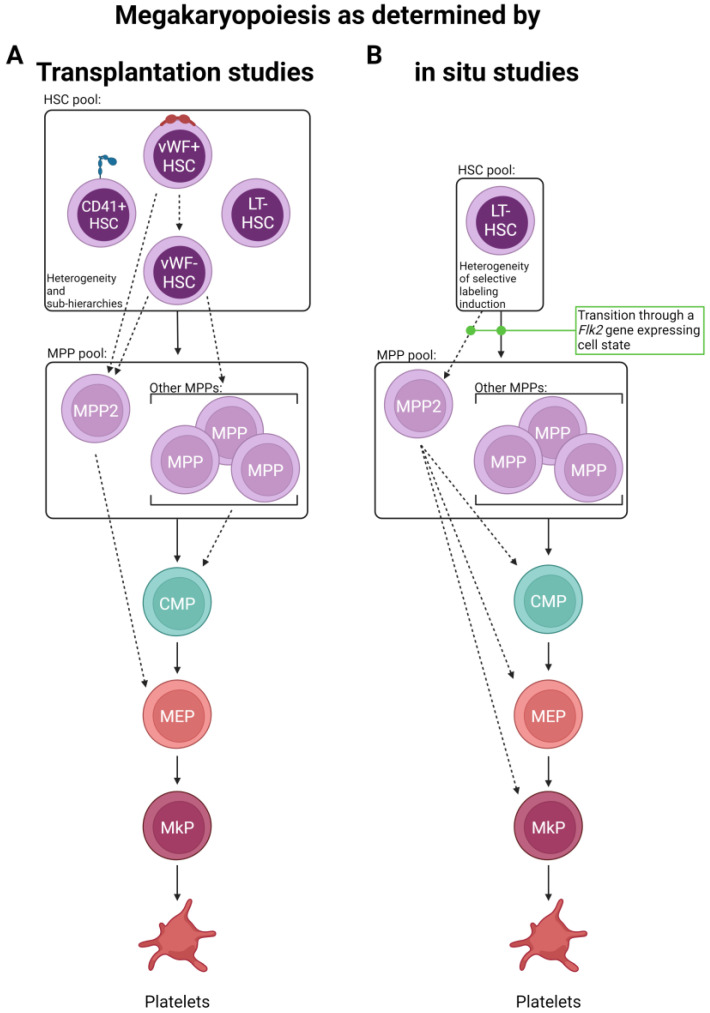
Routes of platelet generation revealed by lineage tracing. Two major lineage tracing methods have been primarily employed to interrogate the route(s) of platelet specification. (**A**) Single and bulk cell transplantation and (**B**) in situ labeling have suggested the possibility of multiple alternative paths of megakaryopoiesis involving the differential use of progenitor cell states. Solid lines indicate “classical” paths, whereas dashed lines represent new and/or expanded differentiation steps elucidated by the studies discussed here. Other cell lineages are omitted for visual clarity.

**Figure 3 cells-13-00704-f003:**
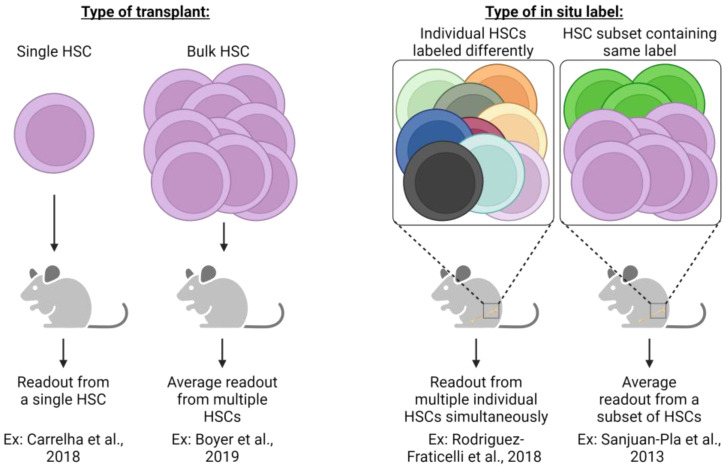
Different experimental strategies require unique interpretations. The experimental approach utilized in lineage tracing studies (transplantation or in situ labeling) necessitates specific interpretation of results and understanding of potential caveats. Single HSC transplants allow an assessment of cellular output from an individual HSC clone, yet readout may be underestimated due to the potential for relatively low-donor-derived chimerism that could be below the method of detection employed. Conversely, bulk HSC transplants significantly improve the detection of donor-derived cells, yet the output of individual HSCs is not possible to assess but is rather the average response of all HSCs transplanted. In situ label induction that uniquely labels each HSC clone allows for the simultaneous assessment of each clone yet may suffer from the same limitations as single-cell HSC transplants (i.e., underestimation of individual HSC contribution due to limits of detection). Similar to bulk HSC transplants, genetic labeling strategies label only a subset of heterogenic HSCs whose measured output is the average of all labeled cells. Both transplantation and in situ labeling also have the potential to skew resulting data and interpretation due to the cell surface phenotype or type of label induction employed. For example, fluorescent genetic reporters may be detectable in platelets, but genetic barcoding approaches are undetectable due to lack of genetic material. If only some heterogeneous HSC clones are assessed, then results can only be understood based on the phenotypic or transcriptomic profile used experimentally [20,68,70,80].

**Figure 4 cells-13-00704-f004:**
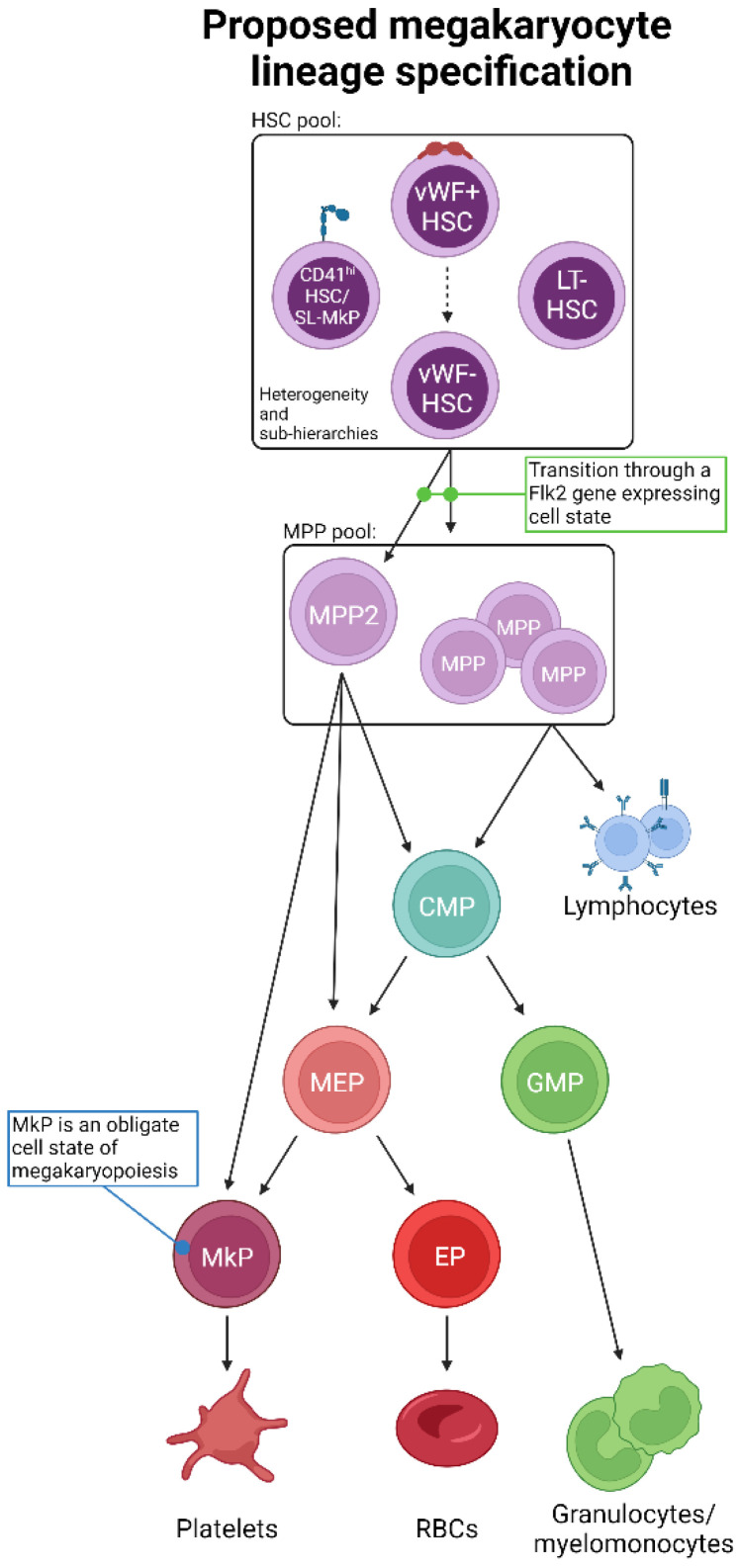
Proposed unification of adult steady-state platelet generation from HSCs, as determined by lineage tracing. Combining the available lineage tracing data, we propose an expanded and unified view of megakaryopoiesis. The phenotypic HSC pool comprises heterogeneous populations likely to be ordered into various sub-hierarchies and may also possess varying degrees of lineage priming, bias, and/or restriction. HSCs then transition to MPPs, including MPP2, which may be a primary subset involved in platelet formation. Importantly, the transition out of the LT-HSC cell state must be accompanied by gene expression of *Flk2*, which is incompatible with a direct HSC-to-platelet path. The “classical” CMP > MEP > MkP differentiation progression may then occur, or specific myeloid progenitor cell states may be bypassed. All possible pathways converge upon the obligate MkP cell state, the maturation of which into megakaryocytes results in eventual platelet production.

**Figure 5 cells-13-00704-f005:**
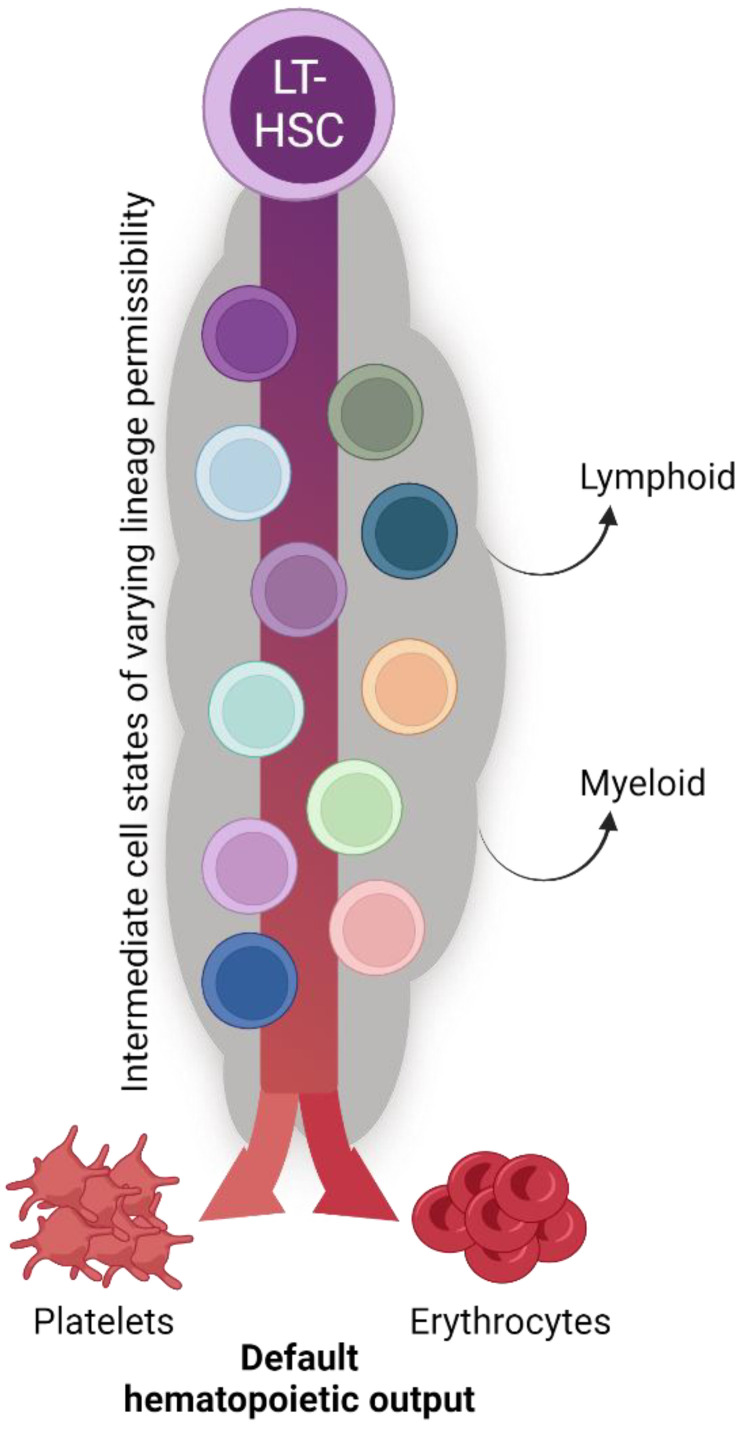
The biological significance of seemingly redundant platelet specification pathways. The collective findings of the studies reviewed herein reveal continual and consistent megakaryocyte lineage generation and platelet production by the hematopoietic system (see also Figure 4). We posit that platelet and erythrocyte production are the default fates of hematopoiesis, with the many shared intermediate progenitor cell states acquiring and/or shifting their differentiation potential to other specific lineages as physiological demand requires. The biological significance of such parallel and redundant paths is to ensure hemostasis and temper the effects of perturbation with respect to platelet output.

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
