# Peer review of "From Hematopoietic Stem Cells to Platelets: Unifying Differentiation Pathways Identified by Lineage Tracing Mouse Models"

_cells, 2024, doi:10.3390/cells13080704_

Round 1
Reviewer 1 Report
Comments and Suggestions for Authors
This is a well-organized comprehensive review article, which cites many papers.
We have learned a lot about the megakaryopoiesis not only the platelet biased/primed or restricted HSC but also other bypass through the CMP and MEP. It’s quite interesting to learn that platelet must go through the MkP stage and all the related mouse models.
It would be very nice, if they could add the heterogeity of megakaryocyte function.
Author Response
Response: We appreciate the Reviewer’s comments and outlook of our manuscript.
As suggested, we have added a section that introduces megakaryocyte functional, transcriptional, and phenotypic heterogeneity and propose a future research direction of assessing if routes of megakaryopoiesis preferentially give rise to specific functional megakaryocyte subtypes.
We are hesitant to expound upon this topic too much as we wish the manuscript to focus primarily on platelet lineage ontogeny as revealed by lineage tracing studies. However, we do believe this to be an important aspect of megakaryopoiesis as recent publications have demonstrated. Thus, we have included this new section as part of the “Conclusions, Outlook, and Open Questions” part of the manuscript.
Reviewer 2 Report
Comments and Suggestions for Authors
The manuscript is a well written, well organized and balanced review, deliberating on the lineage tracing data from the literature as well as from the authors’ laboratory. The authors proposed that platelet differentiation may occur in conjunction with the classical concept of hematopoiesis during steady state. Then they provided evidence from the literature showing that multiple redundancies and/or competencies may co-exist to ensure constant platelet production. At steady state, MPPs transition through the classical hierarchy of CMP>MEP>MkP. Upon acute demand such as hemorrhage, some HSC clones are primed for the megakaryocytic lineage and bypass CMPs and/or MEPs altogether. Some LT-HSCs express megakaryocytic lineage genes but clearly maintain multilineage differentiation potential. They call for further studies to confirm megakaryocyte differentiation pathways outside of steady-state megakaryopoiesis such as during ontogeny and aging, following bleeding, in response to injury and infection. These are the strengths of the review article.
While we have acquired an extensive and detailed understanding of HSC differentiation pathways through studies in murine models, it is still controversial whether this knowledge could be extrapolated to the human system. The classical view based on mouse studies suggests a development from HSCs to multipotent progenitors (MPPs), to lineage-committed progenitors (CMPs) and megakaryocyte-erythroid progenitors (MEPs), evidence from human studies have challenged this linear model. Single-cell RNA sequencing (scRNA-seq) studies of human HSC have shown transcriptional profiles consistent with a direct pathway from HSCs to MEPs. This notion has also been supported by functional assays for the presence of HSCs with megakaryocyte potential, especially upon aging.
In short, evidence from human studies suggests that the pathways governing megakaryopoiesis, or for hematopoietic differentiation at large, may be more flexible and context-dependent. This is consistent with what the authors have discussed for the mouse system. I think the review article could be enhanced by a brief deliberation or discussion on the differences regarding “presence, role, and kinetics of the possible alternate pathways” (in megakaryopoiesis) between humans and mice.
Author Response
Response: We appreciate this perspective and summary of our manuscript.
We agree with the Reviewer’s overall conclusion of the human literature. Clearly, the ability to generate platelets is so critical that multiple, redundant, and/or default programs are likely conserved. Clues from scRNA-seq and in vitro testing certainly are suggestive of a direct pathway from HSCs to MEPs, yet we do not feel there is enough direct evidence in the human system to definitively make that point. The mounting evidence, as the reviewer suggests, makes for a compelling discussion. Thus, we have added a section to the “Conclusions, Outlook, and Open Questions” section that expands upon the existing discussion of human data, its similarity to mice, and indicate that further, targeted studies will be required to assess if megakaryopoiesis discoveries in murine systems are directly applicable to humans.
The enhanced megakaryocyte potential of HSCs upon aging is of significant interest and a point well taken. However, we chose to focus this manuscript on young adult steady-state megakaryopoiesis to lay the framework for overlaying altered/perturbed scenarios (i.e. aging, hemorrhage, infection, malignancy, etc.). We believe that the topic of platelet ontogeny in these states is deserving of their own dedicated review!